# Genome-Wide Identification of the *StPYL* Gene Family and Analysis of the Functional Role of *StPYL9a-like* in Salt Tolerance in Potato (*Solanum tuberosum* L.)

**DOI:** 10.3390/plants14172731

**Published:** 2025-09-02

**Authors:** Chunna Lv, Yuting Bao, Minghao Xu, Ke Deng, Long Zhao, Yihan Zhao, Yifan Zhou, Yuejuan Feng, Fang Wang

**Affiliations:** 1Academy of Agriculture and Forestry Sciences, Qinghai University, Xining 810016, China; lcn2922022@163.com (C.L.); baoyuting0318@163.com (Y.B.); m18697239510@163.com (M.X.); dengker87@163.com (K.D.); longz0322@hainanu.edu.cn (L.Z.); s827506988@126.com (Y.Z.); zhouyifan200109@163.com (Y.Z.); fengyuejuan2025@163.com (Y.F.); 2Laboratory for Research and Utilization of Qinghai Tibet Plateau Germplasm Resources, Qinghai University, Xining 810016, China; 3Qinghai Provincial Key Laboratory of Potato Breeding, Qinghai University, Xining 810016, China

**Keywords:** PYL, potato, PYL gene family, salt stress, RNA sequencing, gene overexpression

## Abstract

PYR/PYL (pyrroloquinoline quinone resistance/PYR1-like) are receptors for abscisic acid (ABA) in plants and play a crucial role in responses to abiotic stress. In this study, we identified 63 members of the StPYL gene family at the tetraploid whole-genome level in potatoes. We analyzed the physicochemical properties of these 63 StPYLs and constructed a phylogenetic tree using *Arabidopsis thaliana* and potato (*Solanum tuberosum* L.) cultivar ‘DM’ as the reference. By examining gene structure, conserved protein motifs, and collinearity, we found that StPYLs are highly conserved throughout evolution. The gene expression heat map under salt stress revealed that 57 StPYL genes are involved in the salt stress response. Among them, the expression level of *StPYL9a-like* changed significantly under salt stress. Through genetic transformation, we observed that overexpression of *StPYL9a-like* enhanced the growth and survival of potato plants under salt stress compared to the wild type. The contents of proline (Pro), superoxide dismutase (SOD), and chlorophyll in the leaves of overexpressing plants increased, while malondialdehyde (MDA) levels decreased. This suggests that *StPYL9a-like* positively regulates salt tolerance by affecting antioxidant enzyme activity and osmotic adjustment substances in potatoes. Subcellular localization demonstrated that *StPYL9a-like* is localized in the nucleus. This study provides a reference for the functional research of PYLs in potatoes, offers a basis for screening potato genes related to salt stress, and lays a foundation for developing salt-tolerant potato varieties.

## 1. Introduction

Abscisic acid (ABA) is a plant hormone with a chemical structure of C_15_H_20_O_4_ [1]. ABA plays a key role in plant growth and development and plant stress responses [2,3]. In the ABA signaling cascade, PYR/PYL receptors serve as pivotal molecular switches that directly interact with the phytohormone to activate downstream signaling cascades, ultimately mediating ABA’s characteristic physiological responses in plants [4].

At present, PYL family genes have been identified in many plants. For example, 14 *PYL* genes with highly conserved amino acid sequences have been identified in *Arabidopsis thaliana* [5]. In rice (*Oryza sativa*) [6], grape (*Vitis vinifera*) [7], tomato (*Lycopersicon esculentum*) [8], rubber tree (*Hevea brasiliensis*) [9], and cotton (*Anemone vitifolia*) [10], 13, 8, 14, 21, and 27 *PYL* genes were identified, respectively. Moreover, PYR/PYL receptors have been widely proven to be closely related to stress physiology, and the functions of some *PYL* genes have been successfully verified. For instance, overexpression of *AtPYL4* enhances drought tolerance in *Arabidopsis thaliana* [11], while *AtPYL8* and *AtPYL9* play important roles in the growth of its lateral roots [12]. The overexpression of *OsPYL5* in rice can enhance the tolerance of rice to drought and salt stress [13], and the overexpression of *OsPYL10* has the potential to improve the drought resistance and cold resistance of indica rice [14]. The overexpression of *ZmPYL8*, *ZmPYL9,* and *ZmPYL12* in maize (*Zea mays*) can enhance the cold resistance [15]. In wheat, *TbPYL2, TbPYL5,* and *TbPYL12* were found to potentially play a role in cold stress [16], and *BnPYR1-3*, *BnPYL1-2,* and *BnPYL7-2* were found in rapeseed to respond to drought, high temperature, and salt stress [17]. Xiaoxia Jia et al. [18] conducted an in-depth analysis of the genome of potato ‘DM-v 6.1’ and identified 17 StPYL genes. The physicochemical properties, evolutionary relationships, gene structures, and expression patterns of these 17 StPYL gene family members under different biotic and abiotic stress conditions were analyzed. The results showed that PYL family members in potato exhibited obvious expression specificity in different tissues (root/stem/leaf etc.). The expression of *StPYL1* was upregulated under treatments with exogenous hormones (such as BAP, ABA, and IAA) and abiotic stresses (such as high temperature, high salinity, and drought), indicating its involvement in the regulation of multiple stress responses. Other StPYL members showed functional differentiation and specific responses to different stresses. Protein–protein interaction and microRNA network analyses revealed that the StPYL family participates in ABA signaling. RT-qPCR validation showed that *StPYL6* was consistently upregulated under drought, salinity, and *Phytophthora* infection. While *StPYL11* exhibited progressive transcriptional suppression under prolonged stress, some members were coordinately downregulated under both abiotic and biotic stresses [19]. *StPYL16* showed a significant response to drought stress: drought enhanced the activation activity of the *StPYL16* promoter on the reporter gene, and transient stable expression of *StPYL16* in tobacco improved the drought resistance of transgenic plants [20]. At present, the whole genome sequencing of tetraploid potato ‘Qingshu 9’ has been completed [21], which lays a foundation for studying evolution and functional analysis of potato gene family systems using bioinformatics methods.

Potato is an important global food crop with high economic benefits and a wide planting range. Soil salinity is one of the major and widespread challenges in the recent era that hinders global food security and environmental sustainability. Worsening the situation, the harmful impacts of climate change accelerate the development of soil salinity, potentially spreading the problem in the near future to currently unaffected regions [22]. With the aggravation of soil salinization, salt stress has become one of the important environmental factors restricting the growth and production of potatoes [23]. Therefore, it is of great significance to improve the salt tolerance of potatoes. In recent years, there has been a growing body of research on the salt tolerance mechanisms of potatoes. For example, GC-MS-based analysis of metabolite changes in potato cultivars ‘Spunta’ and ‘BARI-401’ revealed that potatoes cope with salt stress through activating antioxidant systems, adjusting metabolite compositions, and remodeling metabolic pathways [24]. A comparison of the physiological responses of two potato cultivars (‘Longshu 5’ and ‘Qingshu 9’) to salt stress revealed that potatoes adapt to salt stress by regulating physiological substances such as soluble proteins and antioxidant enzymes. Moreover, strong Na^+^ efflux, K^+^ homeostasis maintenance, and high Ca^2+^ uptake are key mechanisms underlying the salt tolerance of ‘Longshu 5’ [25].

Although the biological characteristics of the *PYL* gene family have been explored in several species, studies on tetraploid potatoes remain limited. Therefore, the aim of this study is to identify the *PYL* gene family in the tetraploid potato variety ‘Qingshu 9’, determine its key functions, and analyze its evolution. To evaluate the role of *StPYL* in regulating salt tolerance, we assessed the expression changes of *PYL* genes under salt stress and investigated the role of *StPYL9a-like* genes in salt stress using genetic transformation studies.

## 2. Results

### 2.1. Identification and Phylogenetic Analysis of PYL Gene Family in ‘Qingshu 9’ Potato

This study identified 63 PYL family genes at the whole genome level of the tetraploid potato variety ‘Qingshu 9’. Based on their chromosomal positions, they were sequentially named *StPYL*1 to *StPYL*62, as well as *StPYL9a-like* (Appendix A). The amino acid sequences of the *StPYL* gene family range from 63 to 1033 amino acids, indicating significant differences in protein sequences among family members. The isoelectric points range from 4.71 to 9.01, with 49 acidic proteins. The molecular weight ranges from 6922.17 to 113,128.52, with *StPYL*35 having the largest molecular weight. The stability coefficients of the proteins range from 25.36 to 58.01, with the instability coefficient of unstable proteins being 41%. The hydrophobicity coefficients range from 0.674 to 0.459, and the lipophilicity coefficients range from 77.29 to 148.57. Subcellular localization predictions indicate that PYL proteins are primarily located in the cytoplasm and nucleus, with less presence in chloroplasts, cytoskeleton, mitochondria, extracellular matrix, and endoplasmic reticulum.

To analyze the evolutionary relationships of the PYL gene family members in *Solanum tuberosum*, a phylogenetic tree was constructed using 14 *Arabidopsis thaliana*, 17 *Solanum tuberosum (DM)* [19], and 63 *Solanum tuberosum (Q9)* PYL proteins (Figure 1). The 63 *Solanum tuberosum* PYL proteins were divided into four subfamilies (Group I–Group IV), with Group I–Group IV containing 2, 16,15, and 30 *Solanum tuberosum* (Q9) PYL genes, respectively. Among them, the 30 PYL genes in Group IV of *Solanum tuberosum* (Q9) are far away from other PYL genes. Whether they are specific subfamilies of Solanum tuberosum (Q9) needs further verification. Among them, the 30 *Solanum tuberosum* (Q9) PYL genes in Group IV are not classified into the three *Arabidopsis thaliana* subfamilies, indicating they are *Solanum tuberosum*-specific genes. The phylogenetic analysis revealed several closely related PYL genes between *Solanum tuberosum* and *Arabidopsis thaliana*, suggesting they may have similar functions.

### 2.2. Distribution of Potato StPYL Family Members on Chromosomes and Analysis of Gene Structure and Protein Conservation Motifs

The 63 StPYL genes are unevenly distributed across the 30 chromosomes of 48 (12 × 4) tetraploid potato (Figure 2). The highest density of StPYL genes was observed on Chr04_A3 and Chr01_A1, with five PYL genes clustered at either end of these chromosomes. *StPYL9a-like* is located at the end of Chr08_A1. These results indicate that potato PYL genes are primarily located in the terminal regions of chromosomes, and the number of genes contained in a chromosome is not related to their length.

### 2.3. StPYL Collinearity Analysis of PYL in Families and Other Species

To further investigate the phylogenetic relationships of the PYL gene family in *Solanum tuberosum*, this study constructed a comparative homology map of the tetraploid *Solanum tuberosum* with *Arabidopsis thaliana* and tomato. The homology between species provides new insights into the evolution and function of gene families. Therefore, we analyzed the collinearity of PYL genes in *Solanum tuberosum* with *Arabidopsis thaliana* and *Solanum lycopersicum* (Figure 3A). The collinearity analysis revealed differences and similarities among the PYL genes in *Solanum tuberosum*, *Arabidopsis thaliana*, and *Solanum lycopersicum*. We conducted collinearity analyses on *Solanum tuberosum*, *Arabidopsis thaliana*, and *Solanum lycopersicum* (Figure 3A). We found that there was a duplication relationship between *Solanum tuberosum* and *Arabidopsis thaliana*, but the collinearity between *Solanum tuberosum* and dicotyledonous *Arabidopsis thaliana* is lower. Notably, one tomato PYL gene has collinearity with multiple potato PYL genes. In addition, we analyzed the colinear relationship between DM and Q9 four groups of chromosomes (Appendix A). These results indicate that *StPYL* exhibits strong conservation during evolution, with a more complex evolutionary process in *Solanum tuberosum*.

### 2.4. Analysis of Intron–Exon Structure and Conserved Motifs in PYL Gene Family

Intron–exon structure analysis shows that the PYL gene family contains multiple introns and exons. Overall, most PYL genes in the same subfamily have similar genetic structures. An analysis of the structures of 63 *StPYL* genes revealed structural variations (Figure 3E). Genes on the same branch of the phylogenetic tree show identical or similar genetic structures. In *StPYL*, introns and exons are alternately arranged in the complete gene sequence. Furthermore, to further explore the structural characteristics of PYL protein, we used the MEME tool to study the divergence of conserved motifs. As a result, 10 conserved motifs were identified and renamed as motif 1–motif 10. An examination of the conserved motifs of StPYL proteins (Figure 3C) indicates that closely related proteins have highly similar motifs. Notably, *StPYL*24, *StPYL*26, *StPYL*45, *StPYL*46, *StPYL*47, *StPYL*54, and *StPYL*52 contain only one motif, which may be due to the loss of bases during tandem duplication. However, most other StPYL proteins contain eight motifs, with the majority having one, two, three, five, six, seven, nine, and 10 motif structures.

### 2.5. Prediction of Cis-Acting Elements in StPYLs

Given the critical role of cis-acting elements in gene regulation, we analyzed the promoter region of StPYLs genes. This analysis revealed numerous light-responsive elements and hormone-responsive elements (specifically for jasmonic acid and abscisic acid) (Figure 3D). We studied the distribution of six cis-regulatory elements under non-biological stress in the StPYL gene promoter region (Appendix A). The ABRE motif was the most abundant, totaling 134, followed by GT1 (*n* = 75), CGTCA motif (*n* = 59), LTR (*n* = 40), TC-rich repeats (*n* = 29), and MBS (*n* = 19). Among these, *StPYL9a-like* includes ABRE, GT1, and the CGTCA motif.

### 2.6. Analysis of StPYLs Candidate Genes Under Salt Stress

Analysis of the expression profiles of StPYLs in leaves and roots at different time points using potato transcriptome data shows that most StPYLs are expressed in tissues and organs (Figure 4). The differences in expression patterns are statistically significant. Heat maps of these genes under salt stress conditions show that most *StPYLs* participate in the response to salt stress. Compared with the control group, 63 PYL expression processing groups showed 29, 22, 25, and 28 differentially upregulated genes in leaves at 3, 6, 12, 24, and 48 h and 22, 14, 27, 30, and 30 differentially upregulated genes in roots at 3, 6, 12, 24, and 48 h, respectively. Among them, the expression of *StPYL9a-like* shows a regular pattern with increasing salt stress. The expression levels of *StPYL9a-like* at 3, 6, 12, 24, and 48 h of salt stress show significant changes and they increased significantly most of the time. At 3 h of stress, the expression level of *StPYL9a-like* increased. At 6 h of stress, it decreases compared to 3 h. From 6 h to 48 h, as the duration of stress increased, the expression level gradually rose. These results indicate that *StPYL9a-like* plays a crucial role in salt stress and that it plays a positive regulatory role.

### 2.7. Subcellular Localization Analysis of StPYL9a-like

In order to verify the function of *StPYL9a-like*, we performed subcellular localization and found that the GFP signal of *StPYL9a-like* was co-localized with the nuclear localization signal (Figure 5), indicating that *StPYL9a-like* is localized in the nucleus.

### 2.8. Overexpression of StPYL9a-like Improves the Growth Traits of ‘Qingshu 9′ Potato

The overexpression (OE) plants of *StPYL9a-like* showed a significant growth phenotype advantage compared with the wild type (W) (Figure 6A), and their plant height, stem diameter, and the number of leaves were significantly higher than those of the wild type (Figure 6B–D). Among them, the plant height of overexpression line 1 (OE-1) and overexpression line 2 (OE-2) were 1.24 times and 1.09 times that of W, respectively (Figure 6B). The stem diameter of OE-1 and OE-2 were 1.25 times and 1.16 times that of W, respectively (Figure 6C). The number of leaves of OE-1 and OE-2 were 1.89 times and 1.25 times that of W, respectively (Figure 6D). The determination of physiological indexes showed that the Pro content, SOD activity, and chlorophyll content of transgenic lines were significantly higher than wild type, and the MDA content was significantly lower than wild type (Figure 6E–H). Among them, the Pro contents of OE-1 and OE-2 were 1.39 times and 1.37 times that of W, respectively (Figure 6E). The SOD activities of OE-1 and OE-2 were 1.02 times and 1.03 times that of W, respectively (Figure 6F), and the chlorophyll contents of OE-1 and OE-2 were 1.02 times and 1.04 times that of W, respectively (Figure 6G). The MDA content of W was 1.37 times that of OE-1 and 1.18 times that of OE-2, respectively (Figure 6H). The relative expression levels of *StPYL9a-like* gene in OE-1 and OE-2 were 1.49 and 1.45 times that of W, respectively (Figure 6I).

### 2.9. Overexpression of StPYL9a-like Enhanced the Salt Tolerance of Potato

To explore the function of *StPYL9a-like* under salt stress, we treated W, OE-1, and OE-2 with 300 mM NaCl solution. After 2 days, it was found that the leaves of WT wilted and showed partial necrosis, while the leaves of OE-1 and OE-2 only wilted slightly (Figure 7A). This observation shows that the growth potential of WT plants under salt stress is significantly weaker than that of OE-1 and OE-2. To further understand the salt tolerance of transgenic lines, we studied the changes in the physiological indicators of potatoes. After treatment with NaCl solution, the content of Pro and the activity of ROS scavenging enzyme SOD in WT and transgenic potato plants increased. It was found that the increase in enzyme activity in OE-1 was significantly higher than that in WT (Figure 7B,C), indicating that the transgenic plants have stronger osmotic regulation and ROS clearance capabilities. The total chlorophyll, chlorophyll a, and chlorophyll b content of transgenic potatoes under salt stress was higher than that of WT (Figure 7D,E), indicating that transgenic plants could cope with oxidative damage more effectively and maintain the function of the photosynthetic system under salt stress. Furthermore, the MDA content of transgenic potatoes decreased after salt treatment (Figure 7F), indicating that the overexpression of *StPYL9a-like* enhanced the cell membrane’s ability to resist oxidative damage. These results indicate that the overexpression of *StPYL9a-like* enhances the salt tolerance of potatoes. These physiological changes synergistically enhance the ability of transgenic potatoes to adapt to salt stress, thereby maintaining their normal growth and survival capabilities. Under salt stress, the expression level of *StPYL9a-like* in wild type potatoes significantly increased, indicating that *StPYL9a-like* responds to salt stress. The expression level of *StPYL9a-like* in transgenic potatoes under salt stress increased more than the increase in wild type potatoes, suggesting that overexpression of *StPYL9a-like* may enhance the response of potatoes to salt stress (Figure 7G). In addition, when potatoes were continuously and quantitatively watered with 300 mM NaCl for 15 days, the transgenic plants had stronger growth vigor than the wild type, this observation indicates that the *StPYL9a-like* transgenic plants are more salt tolerant than the wild type (Figure 7H). In conclusion, these results indicate that overexpression of *StPYL9a-like* in potatoes can enhance the salt tolerance of potatoes.

## 3. Discussion

Potato is an important horticultural crop; however, salt stress severely restricts its growth and development as well as tuber yield. Therefore, identifying salt-tolerant genes in potato and breeding salt-tolerant potato varieties can not only enhance the resistance of potato to salt stress but also effectively improve the planting efficiency of saline-alkali land. With the completion of genome sequencing of ‘Qingshu 9’, conditions are provided for further study of the tetraploid PYL gene family in potato [21]. The PYL genes play a vital role in the ABA signaling pathway of potatoes, but its role has not been reported in tetraploid potatoes. Therefore, this study identified 63 PYL genes in the tetraploid genome of ‘Qingshu 9’ (Appendix A). Compared with the 14 identified species of *Arabidopsis thaliana* [26,27], tomato [8], and rice [28], the number of StPYL gene family members is larger, and most StPYLs are located in the cytoplasm, which is consistent with the subcellular localization results of AtPTLs reported previously [26]. These genes share a common ancestor and exhibit structural and functional similarities. In terms of quantity, the PYL genes in ‘Qingshu 9’ have undergone significant expansion compared to those in *Solanum tuberosum* and *Arabidopsis thaliana*, which might explain its superior stress resistance. In this study, a phylogenetic tree comparing ‘Qingshu 9’ and *Arabidopsis thaliana* was constructed, revealing that, after diverging from Arabidopsis, potato PYL genes have diversified into a new clade. This expansion of the PYL genes family may be a key factor contributing to the strong stress resistance of potatoes, indicating that PYL genes copy number increase in potatoes is closely associated with potatoes’ stress resistance. Analysis of the gene structures, conserved motifs, and domains revealed that conserved motifs are highly similar and exhibit more consistent distributions within each group, suggesting that although there are variations in PYL gene number among ‘Qingshu 9’ genes, the evolutionary process of their sequences is relatively conservative. With the exception of three genes in group IV, all other subfamily genes have multiple introns, indicating that these genes acquired new introns during evolution, which is consistent with PYL in tomato, rice, maize, and grape [8,29,30]. Promoter region analysis of potato PYL genes also identified a large number of cis-elements related to stress and hormone responses, further highlighting the potential role of PYL genes in stress resistance.

This study revealed the dynamic expression pattern of the *StPYL* gene family under salt stress through transcriptome data analysis. Among them, *StPYL9a-like* showed significant time-specific regulatory characteristics. The rapid upregulation of *StPYL9a-like* after 3 h of salt stress may reflect the early response of plants to salt stress. Existing studies have shown that ABA can bind to PYL, and the complex formed after binding will inhibit the enzymatic activity of protein phosphatase (PP2C), resulting in the activation and release of serine/threonine protein kinase SnRK2. The activated SnRK2 kinase regulates downstream transcription factors or proteins, thereby causing the physiological response of ABA [31,32]. At this stage, the rapid synthesis of ABA may initiate downstream signaling pathways by activating PYL receptors, thereby triggering primary stress resistance responses such as stomatal closure and osmotic regulation [33]. When the plants were under stress for 6 h, its expression briefly declined, indicating that that there might be a negative feedback regulation mechanism. For example, after the ABA signal is overly activated, the receptor abundance is regulated through transcriptional inhibitors (such as miRNAs) to avoid excessive energy consumption or signal disorder [34]. It is notable that as the stress duration extended from 12 to 48 h, the expression of *StPYL9a-like* continued to rise again, implying that it plays an important role in maintaining long-term stress adaptation. This bimodal expression pattern may be related to the early rapid defense against salt stress and the later steady-state reconstruction [35]. Furthermore, the differential expressions of *StPYL9a-like* in roots and leaves may reflect tissue-specific regulatory strategies. For example, the root system coordinates ion segmentation (such as vacuole Na^+^ isolation) by enhancing ABA perception [36], while the leaves focus on stomatal regulation and photosynthetic protection [37].

When exposed to salt stress, plants initiate a series of morphological, physiological, biochemical, and genetic changes to regulate their adaptation to internal and external environments [38,39]. In this study, the growth of OE plants was significantly better under both conventional growth conditions and salt stress, suggesting that StPYL9a-like may help plants resist salt stress to some extent. Proline (Pro) is a widely reported osmotic regulator and signaling molecule that accumulates in plants under salt and drought stress. It may protect plants from such stresses primarily through maintaining osmotic balance, aiding reactive oxygen species (ROS) scavenging, regulating antioxidant metabolites, and modulating components of key antioxidant defense enzymes [40]. In this study, the increased Pro content in transgenic lines under salt stress implies that overexpression of StPYL9a-like might enhance osmotic regulation, ROS clearance, and antioxidant defense capabilities in potatoes, which could contribute to the observed improvement in salt tolerance [41].

Superoxide dismutase (SOD) is a core enzyme in the plant antioxidant system. By catalyzing the conversion of superoxide anion radicals into hydrogen peroxide and oxygen, it may reduce ROS-induced damage to membrane lipids, proteins, and DNA. Elevated SOD activity is often considered an indicator associated with enhanced plant stress resistance [42]. In this study, the higher SOD activity in salt-stressed transgenic lines suggests that overexpression of StPYL9a-like could be linked to improved salt tolerance in potatoes. Meanwhile, the reduced MDA levels might indicate alleviated cellular membrane damage [43]. Under salt stress, the decrease in MDA content in transgenic plants (in contrast to the increase in wild type plants) suggests that OE plants may have experienced relatively less cell membrane damage, implying that overexpression of StPYL9a-like might enhance cell membrane stability.

The retention of chloroplast structure in salt-stressed plants might be related to efficient ROS scavenging by elevated SOD activity. Additionally, the accumulation of non-enzymatic antioxidants such as Pro could directly neutralize free radicals. Furthermore, the reduction in membrane lipid peroxidation (as indicated by lower MDA levels) may reflect decreased cell membrane damage, which in turn might help maintain the structural integrity of chloroplasts [44].

In addition, compared with the wild type, *GmLecRlk* overexpressing soybean lines had significantly enhanced fresh weight and Pro content as well as reduced MDA content under salt stress. These results show that *GmLecRlk* gene response to salt stress in soybean [45]. Overexpression of *VaPYL4* in Arabidopsis also enhanced plant resistance to salt stress. Physiological analyses showed that transgenic plants had much lower content of MDA and higher POD activity [46]. Our research results are consistent with these studies.

In summary, the results of this study indicate that overexpression of *StPYL9a-like* significantly enhances the salt tolerance of transgenic potatoes under salt stress. Therefore, we conclude that *StPYL9a-like* responds to salt stress and positively regulates it under such conditions, offering potential for improving potatoes’ tolerance to salt stress and aiding in the development of new salt-tolerant potato varieties. In the future, we will further elucidate the detailed regulatory mechanisms of *StPYL9a-like*, including protein phosphorylation and dephosphorylation, protein interactions, and potential protein modifications.

## 4. Materials and Methods

### 4.1. Identification of PYL

The genome sequence of the reference potato genome “Qingshu 9” was obtained [21]. It was obtained from http://bigd.big.ac.cn/ (accessed on 15 November 2024) at BioProject accession PRJCA006096. Using the PYL of *Arabidopsis thaliana* as a reference, we authenticated the identified potato PYL genes. A total of 14 PYL genes were identified in Arabidopsis. The protein sequences for these coding genes were obtained from Tair (https://www.arabidopsis.org/, accessed on 28 June 2023), and multiple sequence alignments were performed using NCBI BLAST+ (V2.11.0) to authenticate the PYL in the 9th chromosome of the purple potato. The E-value < 1 × 10^−5^ was set as a significant threshold. In the Pfam database (http://pfam.xfam.org/browse, accessed on 15 November 2024), the HMM file for the Polyketide_cyc2 domain (PF10604) was downloaded, and using HMMER V3.1b2 (http://www.hmmer.org, accessed on 15 November 2024) software, a Hidden Markov Model (HMM) was constructed. The PYL members predicted by PFAM data were combined with those predicted by BLAST to obtain the PYL of *Solanum tuberosum*. All candidate PYL family gene sequences were confirmed using the NCBI Conserved Domain Database (https://www.ncbi.nlm.nih.gov/Structure/cdd/wrpsb.cgi, accessed on 16 November 2024) and the SMART program (http://smart.embl-heidelberg.de/, accessed on 16 November 2024).

### 4.2. Construction of Potato PYL Gene Family Tree

The protein sequences of PYL gene family in potato Qingshu 9 were compared using muscle tool (v5.1) [47]. An IQ-Tree (V1.6.12) was used to construct the gene tree [48]. The gene tree was also visualized on the ITOL website [49]. We determined the number of amino acids, isoelectric point (pI), and molecular weight (MW) using the ExPASy website (http://web.expasy.org/, accessed on 29 November 2024). We predicted subcellular localization of candidate genes using the YLoc website (https://wolfpsort.hgc.jp/, accessed on 29 November 2024). We used TBtools software (TBtools-II) to locate gene distribution on chromosomes and employed MCScanx(v1.0.0) for replication analysis of gene families to identify gene duplication events.

### 4.3. Analysis of Exon-Intron Structure and Protein Conservation Motifs

The CDS sequence and genomic sequence of PYL genes in ‘Qingshu 9’ were compared using TBtools software (TBtools-II) to analyze the exon–intron structure of the gene [25]. Then, on MEMEsuite website (http://meme-suite.org/, accessed on 30 November 2024), the conserved motifs of candidate potato PYL genes were identified, and 10 motifs were set as restrictions [50].

### 4.4. Collinearity Analysis in PYL

The PYL of *Arabidopsis thaliana*, *Solanum tuberosum* (‘Qingshu 9’), and *Solanum lycopersicum* were analyzed by MCScan (V1.3) software [51]. First, the protein sequences of them were compared using BlastP (V2.10.1+) program with e-value of 1 × 10^5^. Then we used MCScanX to identify collinear blocks with parameter settings of-k: 50, -s: 5, and-m: 25. We used the duplicate_gene_classifier program from MCScanX to classify the copy types of each gene. Information about PYL was extracted from the result file.

### 4.5. Expression Pattern Analysis of PYL

To explore the gene expression patterns of PYL family genes, we generated three Illumina RNA-seq datasets from two organs (leaves and tubers) of *Solanum tuberosum* (‘Qingshu9’). Raw sequencing reads were subjected to quality control using FastQC (v0.11.9) to evaluate base quality scores, GC content, and adapter contamination. Low-quality reads (Phred score < 20). Filtered clean reads were aligned to the Solanum tuberosum reference genome (version Q9, https://ngdc.cncb.ac.cn/gwh/submit/27179/step?status=1, accessed on 30 November 2024) using HISAT2 (v2.1.0) with parameters --dta --max-intronlen 5000 to enable downstream transcript assembly. Transcriptome assembly and quantification were performed using StringTie (v1.3.3b) with the --merge option to generate a unified transcriptome annotation, and FPKM (fragments per kilobase of transcript per million mapped reads) values were calculated for each gene using the -e -B flags to quantify expression levels [52]. The expression levels of 63 gene family members are shown in the attached table(Appendix A).

### 4.6. Subcellular Localization of StPYL9a-like

The gene cloning vector was detected with PCR using specific primers to amplify the CDS sequence of the StPYL9a-like gene without stop codon. Next, this was ligated into the 35S GFP-1300 RUAN′ vector at the BamHI and SpeI sites to construct the 35S GFP-1300 RUAN′-StPYL9a-like recombinant vector. First, the recombinant vector was transferred into the *Agrobacterium tumefaciens* GV3101 strain. Subsequently, 5 mL of the recombinant plasmid 35S GFP-1300 RUAN′-*StPYL9a-like* bacterial culture was transferred into a 50 mL centrifuge tube, and 50 mL of LB liquid medium supplemented with kanamycin and rifampicin (Kan+rif) was added for expansion culture, which was incubated at 28 °C with shaking at 200 rpm until the OD600 value reached 0.8~1. The bacterial culture was then resuspended in MES buffer and diluted to an OD600 value of 0.6~0.8. For the injection solution preparation, acetosyringone (AS) was added at a ratio of 1:1000 (with a final concentration of 100 µmol) and mixed thoroughly. This was followed by mixing the marker and recombinant plasmid bacterial culture at a ratio of 1:3, which was then set aside for use. Five-to-six-week-old *Nicotiana benthamiana* plants with uniform growth status were selected and irradiated under incandescent light for 2~3 h to ensure stomatal opening. Finally, a 5 mL syringe was used to draw the injection solution, which was then injected into tobacco leaves while avoiding the main veins. The injected tobacco plants were subjected to dark culture for 24 h, after which they were transferred to a light incubator for further cultivation for 48~72 h [53]. Observations were conducted using a laser confocal microscope (TCS-SP8; LEICA Microsystems, Wetzlar, Germany) equipped with a LEICA DFC9000 GT camera. For imaging, the epidermal cells were excited with a 514 nm laser, and the emission signals were collected within the wavelength range of 524~574 nm.

### 4.7. Plant Materials and Growth Conditions

Potato (*Solanum tuberosum* L.) cultivar ‘Qingshu9’ (Qinghai Academy of Agricultural and Forestry Sciences, China) plantlets were cut into stem segments in vitro with one or two leaves and were inoculated in Murashige and Skoog (MS) solid medium [54]. The samples were placed in a light incubator at a temperature of 23 °C with 16 h of light and 8 h of darkness and grown for about 21 d for subsequent experiments.

After 21 days of growth, tissue culture seedlings with relatively uniform growth status were selected and inoculated onto MS medium containing 0 mmol/L NaCl (control group) and 300 mmol/L NaCl (treatment group), respectively. There were 6 plants per bottle, with each bottle serving as one replicate, and 3 replicates in total. The plants were arranged uniformly in each medium. Sampling was performed at 3, 6, 12, 24, and 48 h after stress initiation. For both the treatment and control groups, samples from 3 biological replicates were collected separately and pooled as samples for transcriptome analysis.

Next, the Bentham’s tobacco (*Nicotiana Benthamian* L.) seeds were cultivated in pots (5 × 5 cm) supplemented with nutrients soil and vermiculite with a ratio of 1:1, and the moisture content of the soil was maintained (70~75%). These pots were placed in an incubator with a 16 h light (5000 Lx)/8 h dark photoperiod at 23 °C for three to four weeks for growth [55].

### 4.8. Cloning of StPYL9a-like Gene

According to the method described in the instruction manual of the 2× Fast Taq PCR Master Mix amplification kit (Takara Biotech, Beijing, China), PCR amplification was performed using the cDNA after reverse transcription of *Solanum tuberosum* (‘Qingshu 9’) RNA as the template. Primers were designed using SnapGene^®^ 4.2.4, as shown in Appendix A. The sequences of the *StPYL9a-like* gene (ID: XM_006343807.2) was retrieved from the potato database Spud DB (http://spuddb.uga.edu, accessed on 10 November 2023). The coding sequence (CDS) of the *StPYL9a-like* gene was cloned using the cDNA of potato cultivar ‘Qingshu 9’ leaves as a template. The reaction system was as follows: OE-*StPYL9a-like*-F 1 μL, OE-*StPYL9a-like*-R 1 μL, cDNA template 1 μL, 2× Easy Taq PCR Master Mix 10 μL, and ddH_2_O 7 μL. A list of all primer sequences is shown in Appendix A. The samples were pre-denatured at 95 °C for 5 min, and 35 cycles were performed at 95 °C for 20 s, 60 °C for 20 s, 72 °C for 70 s, finally 72 °C for 5 min and stored at 4 °C.

### 4.9. Real-Time Fluorescent Quantitative PCR Assay

The concentration of cDNA was measured with an Nanodrop one ultra-micro spectrophotometer (Thermo scientific, Beijing, China), and the concentration was diluted to about 200 ng/μL. The qRT-PCR analysis was performed according to the TB Green premix Ex Taq II instructions (Takara, Japan). The relative mRNA expression of the target gene was determined using a qRT-PCR instrument (Roche LightCycler^®^ 96, Beijing, China), and the reaction system was 10 μL of TB Green premix Ex Taq II, 0.4 μL of each forward and reverse primers, 2 μL of cDNA template (200 ng/μL), and 7.2 μL of nuclease-free water. The total reaction system was 20 μL. Furthermore, the reaction conditions were 95 °C for 30 s, followed by 40 cycles of 95 °C for 10 s and 57 °C for 32 s, then 95 °C for 60 s, 57 °C for 60 s, and 55 °C for 10 s. *St actin* (GenBank No. X83206) [56] was used as the standard reference gene. Three biological replicates and three technical replicates were performed for each experiment, and relative gene expression was calculated using the 2^−ΔΔCt^ method [57]. For qRT-PCR analysis, the specific primers were designed using NCBI’s primer-BLAST online analysis tool (https://www.ncbi.nlm.nih.gov/tools/primer-blast/, accedded on 29 November 2023). The list of all primer sequences are shown in Appendix A.

### 4.10. Genetic Transformation of Potato and Identification of Transgenic Plants

The coding sequence of the *StPYL9a-like* gene was amplified according to the designed specific primers (Appendix A) and inserted into pJAM1502 vector using Gateway technology. The primary method involves ligating the CDS fragment of the *StPYL9a-like* gene into the pDONR207 entry vector using a BP reaction, followed by extraction of the pDONR207-*StPYL9a-like* plasmid. Subsequently, the pDONR207-*StPYL9a-like* plasmid is recombined with the pJAM1502 expression vector through an LR reaction to generate the pJAM1502-StPYL9a recombinant vector. Detailed procedures refer to the instructions of the Gateway kit (Gateway^®^ BP Clonase™ Enzyme Mix II, Gateway^®^ LR Clonase™ Enzyme Mix II, Invitrogen Biotech, Beijing, China). Agrobacterium suspensions containing recombinant plasmids were cultured in LB liquid medium (containing 50 μg/mL Rif, 50 μg/mL Kan) and activated to OD600 = 0.8. Then, the *Solanum tuberosum* (Qingshu9) was transformed by Agrobacterium mediated stem method [58,59]. The overexpression lines of transgenic *StPYL9a-like* were obtained through antibiotic screening and PCR confirmation of transgenic seedlings, and further verified using physiological index determination and qRT-PCR. The transgenic plants were cultivated on Murashige and Skoog (MS) medium supplemented with timentin and kanamycin sulfate [55]. The condition for growth is 25 °C under a 16 h light/8 h dark photoperiod with a light intensity of 10,000 Lx.

### 4.11. Salt Stress Treatment of Wild Type and Genetically Modified Potatoes

Transgenic lines and wild type potato plants grown for 30 d in MS medium (pH = 6.1) were planted in pots (12 × 12 cm) and supplemented with soil nutrients and vermiculite (1:1, *v*/*v*). One day before transplantation, each pot was watered with 200 mL of deionized water to fully moisten the substrate. For WT, OE-1, and OE-2, 2 pots were transplanted for both the treatment group and the control group, with 6 plants per pot. After transplantation, the plants were placed in an artificial climate chamber for a 15-day acclimatization period. The plantlets were cultivated in a climate chamber with a light intensity of 10,000 Lx, a temperature of 24 ± 2 °C, and a photoperiod of 16 h of light and 8 h of darkness. During this period, 200 mL of deionized water was applied every 3 days to maintain stable substrate moisture, avoiding drought or waterlogging. The growth status of the plants was observed daily, and withered or abnormal individuals were removed promptly to ensure uniform plant growth before treatment initiation. Exactly 17.53 g of NaCl (analytical grade) was weighed, dissolved in deionized water, and volumetrically adjusted to 1 L. The solution was stirred until completely dissolved and prepared fresh for use. The treatment group was watered with 500 mL of 300 mmol/L NaCl solution at 9:00 AM daily, ensuring uniform penetration of the solution to the bottom of the substrate (excess solution could drain through the drainage holes at the bottom of the pot to avoid salt accumulation). The control group received 500 mL deionized water at the same time daily under identical environmental conditions. Throughout the stress period, samples were collected daily at 3:00 PM by harvesting the third to fourth functional leaves (counting from the apical meristem), excising petioles, rinsing leaf surfaces briefly with deionized water to remove dust, and blotting dry with absorbent paper. Each sample was divided into two portions: one portion of fresh tissue was used for the determination of physiological indices, while the other portion (leaf segments) was snap-frozen in liquid nitrogen for 5 min and then stored at −80 °C for subsequent RNA extraction/analysis.

### 4.12. Measurement of Growth and Physiological Indices

The plant height and stem diameter of potato were measured before and after salt stress treatment using a tape measure (precision 0.1 cm) and a vernier caliper (precision 0.1 mm), respectively. The number of leaves was manually counted. The contents of Pro [60] and MDA [61], SOD activity [62], and chlorophyll content [63] were determined using assay kits (Nanjing Jiancheng Bioengineering Institute, Nanjing, China). Specifically, the third and fourth functional leaves of potato were collected, and relevant physiological indexes of these leaves were measured according to the kit instructions.

### 4.13. Data Analysis

IBM SPSS Statistics 26 was used for significance analysis. Significant differences between the samples were considered when *p* < 0.05, and Microsoft Excel 2016 was used for data organization and some preliminary analyses.

## 5. Conclusions

In this study, 63 *StPYL* genes were identified in the tetraploid cultivated potato variety ‘Qingshu 9’ and were classified into 4 subfamilies. These genes exhibit evolutionary conservation and include potato-specific members. Their gene structures, conserved motifs, and stress-responsive cis-elements (such as ABRE) in the promoter regions indicate their potential involvement in stress regulation. Under salt stress, 57 *StPYL* genes are responsive, among which *StPYL9a-like* shows dynamic expression changes and is localized in the nucleus. Genetic transformation experiments demonstrated that overexpression of *StPYL9a-like* enhances the growth and survival ability of potatoes under salt stress by increasing the contents of proline, superoxide dismutase (SOD), and chlorophyll and decreasing the content of malondialdehyde (MDA) in leaves, confirming its positive regulation of potato salt tolerance. In conclusion, this study provides a theoretical basis for the potential application of *PYLs* in the genetic improvement of potato salt tolerance, which is of great significance for potato salt-tolerant breeding.

## Figures and Tables

**Figure 1 plants-14-02731-f001:**
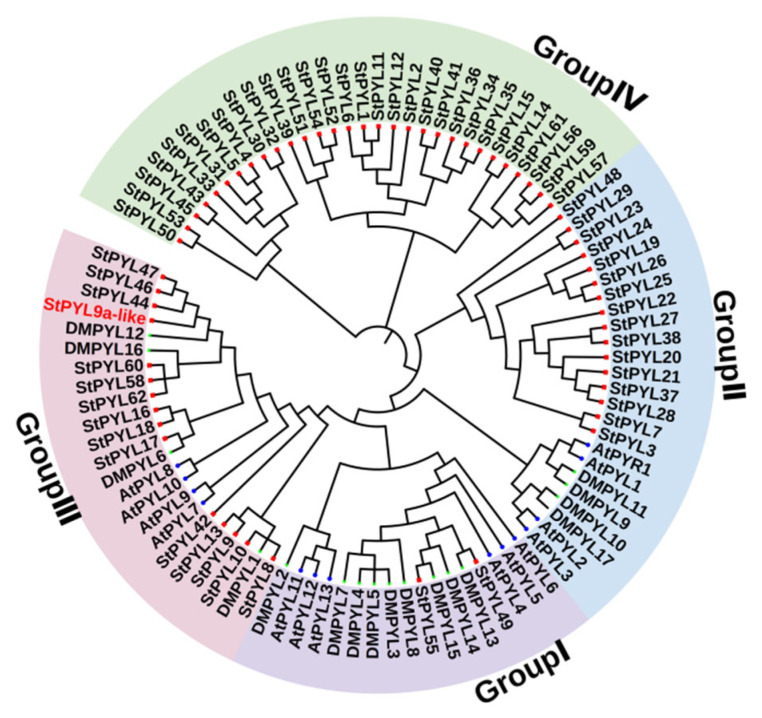
Phylogenetic tree of *Solanum tuberosum* (‘Q9’); *Solanum tuberosum* (‘DM’), and *Arabidopsis thaliana* (At) PYL family members. The red square and blue circle represent the PYL protein of *Solanum tuberosum* and *Arabidopsis thaliana*, respectively. The phylogenetic tree was constructed using the neighborhood connection method (NJ) and repeated 1000 times.

**Figure 2 plants-14-02731-f002:**
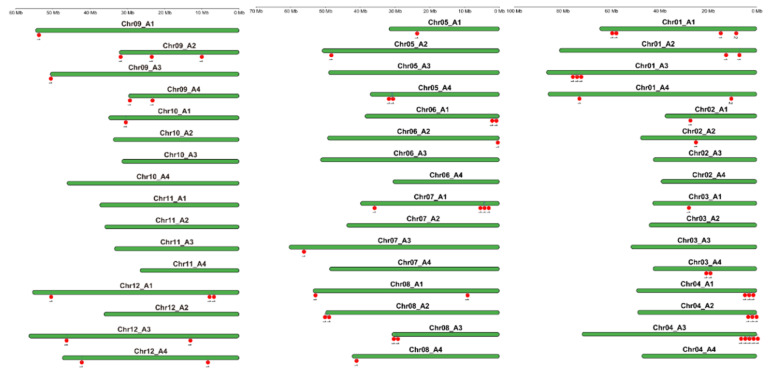
The location and distribution of 63 members of the *StPYL* gene family on the 12 × 4 chromosomes of potato. Green represents each chromosome, and the scale on the left indicates the length of potato chromosome. Red represents the location of each gene.

**Figure 3 plants-14-02731-f003:**
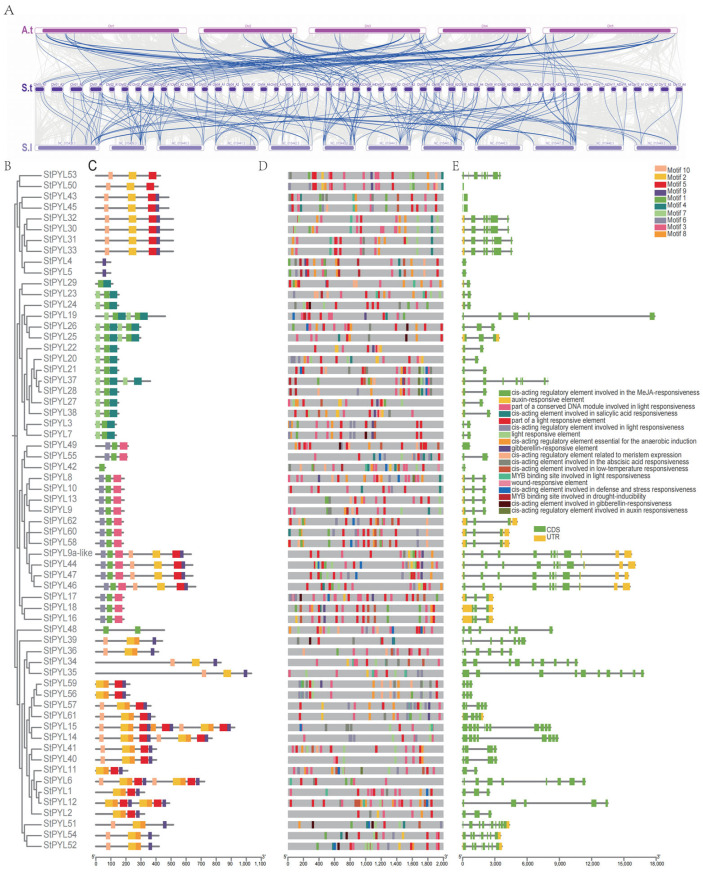
Collinearity analyses of the PYL family genes and conserved motifs, structures, structural domains of StPYLs. (**A**) Homology analysis of PYL family genes in different species. Top: *Arabidopsis thaliana* (A.t); middle: *Solanum tuberosum* (S.t); bottom: *Solanum lycopersicum* (S.l). The numbers on the horizontal line represent the chromosomes of each species. Blue lines represent the collinearity of PYLs. (**B**) Phylogenetic tree of *Solanum tuberosum* (St) of PYLs. The phylogenetic tree was constructed using NJ method with MEGA 6.0. (**C**) Distribution of conserved motifs in PYL protein. Patterns 1–10 are represented by boxes of different colors. (**D**) Cis-elements in the promoter region of *StPYL* genes. Various color bars represent different cis-elements. (**E**) Gene structures of 63 PYL genes in *Solanum tuberosum*. Introns and coding sequences are indicated by black and yellow lines, respectively, with green representing UTRs.

**Figure 4 plants-14-02731-f004:**
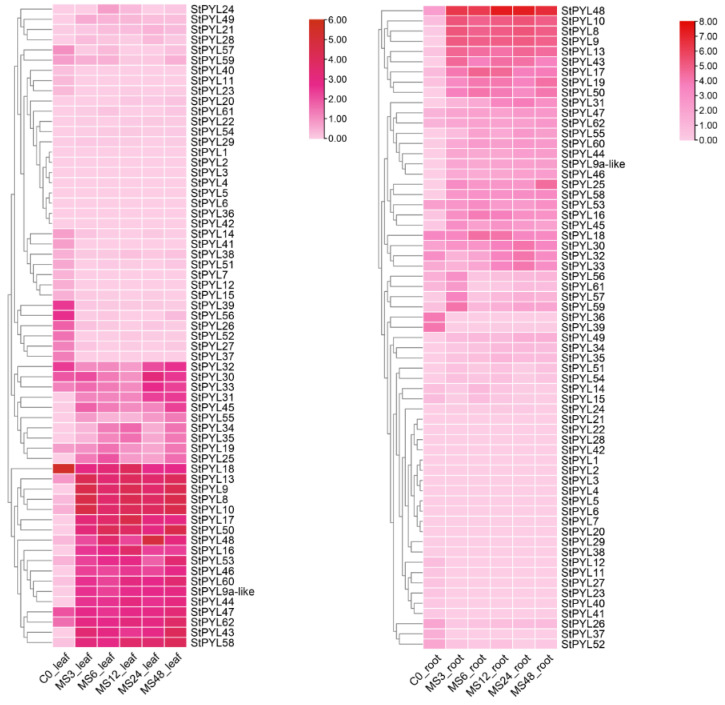
Expression patterns of StPYL genes in response to drought stress (salt treatment) at 0 h, 3 h, 6 h, 12 h, 24 h, and 48 h. Colors in the heatmap represent gene transcript levels, as indicated by the key bar to the right of the figure.

**Figure 5 plants-14-02731-f005:**
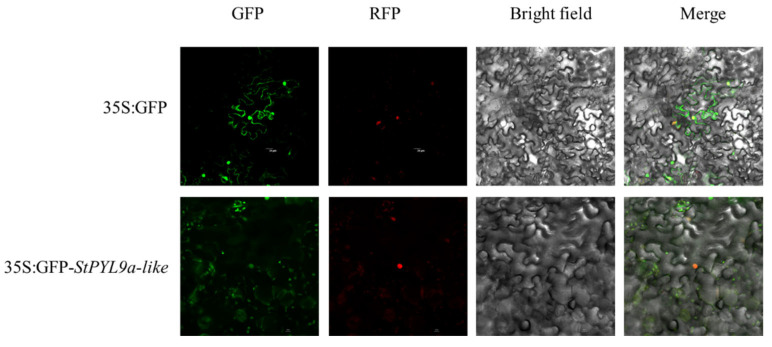
Subcellular localization of StPYL9a-like protein in tobacco epidermis. Green fluorescent protein (GFP), red fluorescent protein (RFP), chlorophyll autofluorescence (Auto), bright-field, and merged images are shown.

**Figure 6 plants-14-02731-f006:**
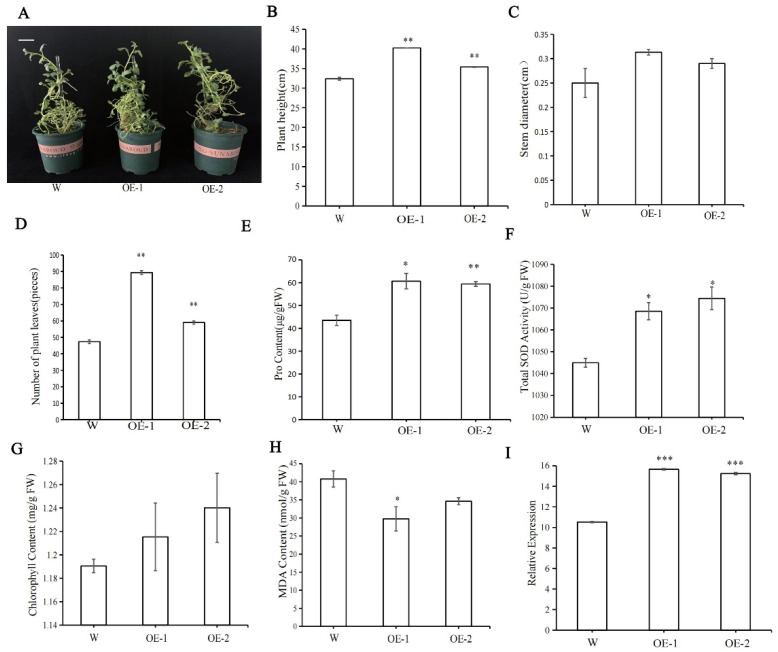
The differences in growth, physiology, and gene expression between transgenic and wild type potatoes. (**A**) Morphological comparison chart of wild type potatoes and genetically modified potatoes. (**B**) Plant height. (**C**) Stem diameter. (**D**) Number of plant leaves. (**E**) Pro content. (**F**) SOD content. (**G**) Chlorophyll content. (**H**) MDA content. (**I**) Relative expression level of genes. Note: statistical significance was determined using a Student’s *t* test (* *p* < 0.05; ** *p* < 0.01; *** *p* < 0.001), data are mean ± SD of 3 independent experiments, *n* = 6 hypocotyl explants for each experiment. Scale bars, 0.5 cm.

**Figure 7 plants-14-02731-f007:**
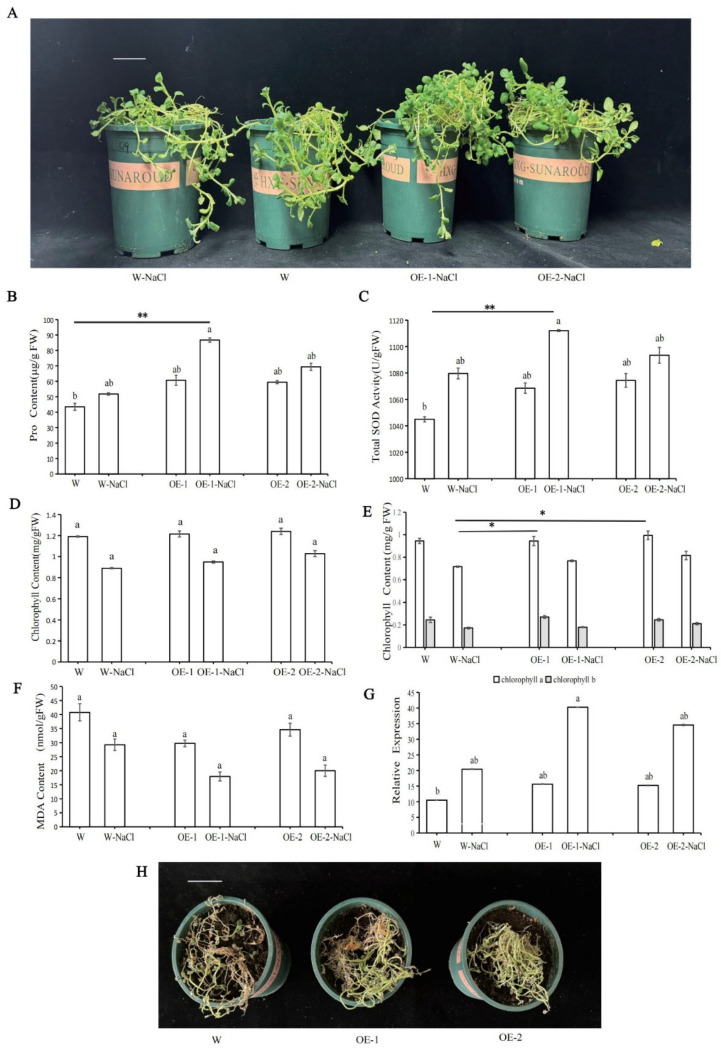
The effects of salt stress on the growth, physiology, and gene expression of transgenic and wild type potatoes. (**A**) Morphological comparison chart of wild type potatoes and genetically modified potatoes under short-term salt stress. (**B**) Pro content. (**C**) SOD content. (**D**) Chlorophyll content. (**E**) Chlorophyll a and b content. (**F**) MDA content. (**G**) Relative expression level of genes. (**H**) Morphological comparison diagrams of potatoes and genetically modified potatoes under long-term salt stress. Note: statistical significance was analyzed with a one sided Kruskal–Wallis test with Bonferroni correction, followed by post hoc Dunn’s test (* *p* < 0.05; ** *p* < 0.01). The same lowercase letters indicate no statistical difference between groups, while different lowercase letters indicate a statistical difference between groups (*p* < 0.05). Data are mean ± SD of 3 independent experiments, *n =* 6 hypocotyl explants for each experiment. Scale bar = 5 cm.

## Data Availability

Data will be made available on request.

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
