# Peer review of "Genome-Wide Identification of the StPYL Gene Family and Analysis of the Functional Role of StPYL9a-like in Salt Tolerance in Potato (Solanum tuberosum L.)"

_plants, 2025, doi:10.3390/plants14172731_

Round 1

Reviewer 1 Report

Comments and Suggestions for Authors

Comments on the Quality of English Language

Reviewer 2 Report

Comments and Suggestions for Authors

The article submitted by Chunna Lv et al., entitled as “Genome-wide identification of StPYL Gene Family and analysis of the functional role of StPYL9a-like in salt tolerance in potatoes (Solanum tuberosum L.)” is comprehensive, informative with novelty of work. In this MS authors discussed the overexpression of StPYL9a-like enhanced the growth and survival of potato under salt stress compared with wild-type plants. The authors presented that the contents of proline, superoxide dismutase, and chlorophyll in the leaves of overexpressing plants increased, and the content of malondialdehyde decreased. This indicates that StPYL9a-like positively regulates the salt tolerance by affecting the antioxidant enzyme activity and osmotic adjustment substance content of potato. They also reported that subcellular localization showed that StPYL9a-like was localized in the nucleus. Moreover, this study provides a reference for the functional research of PYLs in potatoes, offers a basis for screening potato genes related to salt stress, and lays a foundation for the cultivation of salt-tolerant potato varieties. Overall, MS is suitable for the publications. However, I have some comments and suggestions for the betterment of this MS. I would like to recommend minor revision of this MS.

  1. Discussion how salt stress actually affecting the potato yields, growth etc. in the introduction section. At least one mechanisms of action must be added.
  2. How did you apply salt stress to potatoes plants. Please add in material and method section carefully.
  3. Provide proper layout showing treatment plan including control.
  4. In line 437-441 reference must be added
  5. Why did you perform one way ANOVA. Provide justification
  6. Did you measure total chlorophyll or chlorophyll a and b also? If yes, its ok if not why?
  7. Provide standard curve showing measurement and quantification of SOD and POD and also mention the unit of measure.
  8. Which software was used for the functional categorization of reported genes? Provide a list of up and downregulated genes with respect to their ratio.
  9. Provide volcano plot (X and Y axis) showing significantly the up and down regulated genes under salt stress
  10. Provide gene-gene interaction using STRING software
  11. Discussion section should be improved.
  12. Check grammar and typographical errors carefully.
  13. In the discussion section justify your study with reference to other researchers.
  14. If possible, add hydrogen peroxide quantification as it is very important under stress environments.
  15. How your study directly links to the traditional farmers? How your study could benefit farmers add in the conclusion section.

Reviewer 3 Report

Comments and Suggestions for Authors

GENERAL COMMENT

The article presents a relevant topic for alleviating an agronomic problem increasingly present in today's agricultural landscape. It uses standardized methodology and innovative analysis methods, with results consistent with the authors' proposed hypothesis. Therefore, the recommendation is to accept the article for publication as an original article, with some minor recommendations.

SPECIFIC COMMENTS

Introduction: 3rd paragraph, 2nd sentence: "With the aggravation of soil salinization,..." A brief commentary with appropriate bibliographical references explaining why soil salinization is worsening would enrich the work.

The last sentence of the last paragraph of the introduction could be omitted, as it would be implied as part of the conclusion.

Discussion: There is room for enriching and better supporting the discussion of the work by correlating the results obtained with other previous studies on the subject, especially in the first paragraph, in order to lend greater credibility to the results obtained.

Reviewer 4 Report

Comments and Suggestions for Authors

Reviewer report for Plants – MS ID: “Genome-wide identification of StPYL Gene Family and analysis of the functional role of StPYL9a-like in salt tolerance in potato (Solanum tuberosum L.)”

1 · General evaluation & recommendation

The manuscript combines an in-silico inventory (63 StPYLs) with the over-expression of a single member (StPYL9a-like) to suggest improved salt tolerance in potato. While the topic is within the scope of Plants and the study could add value for breeders working on abiotic stress, I identified substantial scientific and presentation flaws (see below).

Recommendation: Major revision (publication is not currently warranted).

2 · Scientific shortcomings

Section               Issue    Evidence           Suggested remedy

Novelty & contextualisation The authors state that the StPYL family “has not been studied in tetraploid potato”, yet at least two recent genome-wide studies (Jia et al. 2022; Gul et al. 2024) already reported StPYL inventories and expression patterns. The manuscript neither compares its 63 genes with the 17 or 57 loci reported earlier, nor clarifies whether the new members are bona-fide genes, alleles or annotation artefacts.                                Provide a detailed cross-reference table (old vs. new IDs), clarify assembly version, and explain why additional loci appear.

Gene identification pipeline Only the PF10604 domain search is mentioned; PYLs also carry the CL0349/CL0457 START-like/Bet V1 folds. No E-value threshold, motif validation or manual curation steps are provided.    Methods section lacks detail.             Include full HMM parameters, remove obvious partial/low-coverage hits, and deposit the FASTA list.

Functional validation design               (i) Single-gene focus with two over-expressing (OE) lines; (ii) only six plants per genotype watered with a very high 300 mM NaCl; (iii) main read-outs are morphological traits and colorimetric kits. No ion content, photosynthetic efficiency, or downstream ABA-signalling markers are shown.      Sample size & treatment: . Replicate number n = 3:            Increase biological replication (≥ 3 independent OE events, ≥ 10 plants each); use realistic salinity (100–150 mM); add Na⁺/K⁺ quantification, stomatal conductance, and expression of canonical ABA-responsive genes.

Statistical analysis     Duncan’s multiple-range test is applied on only three biological replicates; assumptions of normality/homogeneity are not tested, and p-values are not reported.                              Verify assumptions, report exact p-values, and, given small n, use non-parametric tests or increase replicates.

Molecular evidence   No western blot confirms StPYL9a-like accumulation; no PP2C inhibition or SnRK2 activation assays demonstrate functional engagement in ABA signalling.           Entire Results section.                Provide protein-level evidence and at least one downstream biochemical assay.

Bioinformatics interpretation              Authors interpret a large "Group IV potato-specific clade" as evidence of stress-related expansion yet do not analyse selection pressure, duplication mechanisms, or expression patterns of this group.              Phylogeny description (Results 2.1-2.4).     Conduct Ka/Ks, tandem vs. segmental duplication analysis, and show expression heat-maps for all clades.

3 · Linguistic & presentation issues

English quality – frequent grammatical errors, redundant wording (“steam diameter”, “co-linearity”, “saline-alkali conditions”).

Example: “Steam diameter (cm)” instead of stem .

Figure/legend accuracy – Figures 4–6 mix drought (PEG) and salt labels; scale bars lack units in some sub-panels.

Units & symbols – “μm.The” (missing space), “Lx。” (Chinese full-stop) .

Reference style – Several citations lack journal italics and DOI; some references are duplicated (nos. 25 & 41).

Data availability – “Data will be made available on request” does not satisfy the Plants policy; raw RNA-seq, vectors, and phenotypic datasets must be deposited.

4 · Minor comments

Abstract: “63 members at the tetraploid whole-genome level” is ambiguous – specify the cultivar and assembly.

Methods: clarify whether qRT-PCR efficiency was checked and which reference genes were used.

Results 2.6: Figure 4 actually shows drought (PEG) not salt; correct caption.

Discussion: many mechanistic claims (ROS, osmotic adjustment) are speculative without supporting measurements; temper the language.

Please replace all passive tense fragments with active voice where appropriate.

5 · Overall verdict

The study has potential but presently falls short in experimental robustness, statistical power, and comparative context. Addressing the major points above, enriching the datasets, and thoroughly revising the English will be necessary before the manuscript can be reconsidered for publication in Plants.

Reviewer 5 Report

Comments and Suggestions for Authors

The manuscript “Genome-wide identification of StPYL Gene Family and analysis of the functional role of StPYL9a-like in salt tolerance in potato (Solanum tuberosum L.)” by Lv et al. identified 63 StPYL family genes in the tetraploid genome of potato 'Qingshu 9' and analyzed their physicochemical properties, genetic structure, conserved protein motifs, and collinearity. This research revealed that overexpression of the StPYL9a-like gene improves salt tolerance in potato, as evidenced by increased plant growth and survival, as well as increased levels of proline, superoxide dismutase (SOD), and chlorophyll, and decreased malondialdehyde (MDA) content. The results suggest that StPYL9a-like positively regulates salt tolerance by influencing the activity of antioxidant enzymes and the content of osmotic adjustment substances, and its subcellular localization is in the nucleus. This work provides a basis for functional investigation of PYLs in potatoes and contributes to the development of salt-tolerant potato varieties.

While the article presented holds promise, aspects of it would benefit from additional refinement to enhance their robustness and rigor. Specific areas for improvement are outlined below:

Abstract:

Lines 15-17: “We analyzed the physicochemical properties of the 63 StPYLs and constructed a phylogenetic tree with Arabidopsis thaliana”. It would be better to specify that the phylogenetic analysis was performed jointly with the genes of the PYL family of Arabidopsis thaliana.

Lines 19-21: “Among them, the expression level of StPYL9a-like changed significantly at 3,6,12,24 and 48 h under salt stress”. Instead of specifying the times of each treatment, it would be better to leave it as a salt stress experiment.

Introduction:

There are many isolated sentences that could be linked together using connectors. I recommend that authors fix this.

Lines 39-43. These two sentences are closely related, so they should be joined using connectors.

Line 43: “.....genes were respectively identified”. Fix with “.....genes were identified, respectively”.

Lines 43-45: “PYR/PYLs have been widely proven to be closely related to stress physiology. The functions of some PYL genes have been successfully verified”. These two sentences are closely related, so they should be joined using connectors.

Lines 45-47: Arabidopsis thaliana is used very frequently. I recommend restructuring this section.

Line 71: “Phytophthora”. Please, put in italics.

Line 76: “The”. Fix with “the”.

Lines 76-81: This section should be dedicated exclusively to detailing the biological characteristics and economic significance of the "Qingshu 9" variety, thereby emphasizing the rationale for sequencing its genome. The identification of the PYL family should not be introduced here, as it is addressed in the concluding portion of the introduction.

Methodology:

There are many isolated sentences that could be linked together using connectors. I recommend that authors fix this.

Lines 361-362: “Using the PYL of Arabidopsis thaliana as a reference, we authenticated the identified potato PYL”. Why did the authors not use the genes identified in the DM genome as a reference?

Lines 365-366: “....authenticate the PYL in the 9th chromosome of the purple potato”. As I mentioned previously, why were the PYLs identified in the other potato genome not used?

Lines 366-367: “The protein sequences of PYL gene family in potato Qingshu 9 were compared using muscle tool”. Please indicate which evolutionary method and bootstrap were used. I also recommend that authors include the PYL members identified in the DM potato genome in their phylogenetic analysis.

Lines 385-386: “The CDS sequence and genomic sequence of PYL genes in Elymus chinensis were compared using TBtools software to analyze the exon-intron structure of the gene”. Why is that species mentioned in the methodology? I recommend that the authors specify where the CDS sequences were obtained.

Lines 390-391: “The PYL of Arabidopsis thaliana, Solanum tuberosum(Qingshu 9) and Solanum lycopersicum were analyzed by MCScan (V1.3) software”. I recommend that the authors include the potato DM genome in this analysis.

Lines 397-399: “To explore the gene expression patterns of PYL family genes, we generated three Illumina RNA-seq datasets from two organs (leaves and tubers) of Solanum tuberosum(Qingshu9)”. I suggest the authors detail the complete differential expression experiment. This should encompass the experimental design, seedling growth and stress application, sample collection methods, RNA extraction and quantification, and sequencing library preparation. Additionally, a separate paragraph should be dedicated to the bioinformatics analyses, outlining the parameters and criteria used for each program, and how the results were visualized.

Lines 402-408: Please restructure this section because it's not clear. I recommend using connectors.

Lines 432-433: “Fifteen days after transplanting, we watered the potatoes with 500 mL of water containing 300 mmol/L NaCl every day”. The authors do not indicate how often samples were collected once saline stress began.

Results

It is recommended that the authors include information on the PYL genes from the potato DM genome in their analysis. Furthermore, they never justify the designation StPYL9a-like. What is the basis for using this term? For this reason, would using information from the DM genome help in better naming this gene? The authors refer to supplementary tables, but they are not visible. I recommend including the data related to each of the genes identified in this work as supplementary material.

Lines 122-123: “The 63 StPYL genes are unevenly distributed across the 30 chromosomes of 48 potato”. The term “48 potato” is not understood, can you correct it?

Line 241: Figure 6. Statistical significance is not indicated in the figure.

Comments on the Quality of English Language

The manuscript requires revision by a native language specialist to address grammatical errors and improve paragraph structure.

Round 2

Reviewer 1 Report

Comments and Suggestions for Authors

Dear Authors, 

Your manuscript has undergone significant improvements and corrections. Unfortunately, the fluorescent pictures are still of poor quality and they are not suitable for publication. They need to be changed!

9.8.2025

Reviewer 5 Report

Comments and Suggestions for Authors

I have reviewed the resubmission of the manuscript entitled "Genome-wide identification of StPYL Gene Family and analysis of the functional role of StPYL9a-like in salt tolerance in potato(Solanum tuberosum L.)". The authors answered in a satisfactory way the points that I have addressed in the first review. Thus this new version of the manuscript can be accepted for publication.